# Stimulus edges induce orientation tuning in superior colliculus

Yajie Liang [1,6,7], Rongwen Lu [1,7], Katharine Borges[2] & Na Ji [1,2,3,4,5] ✉

Orientation columns exist in the primary visual cortex (V1) of cat and primates but not mouse. Intriguingly, some recent studies reported the presence of orientation and direction columns in the mouse superficial superior colliculus (sSC), while others reported a lack of columnar organization therein. Using in vivo calcium imaging of sSC in the awake mouse brain, we found that the presence of columns is highly stimulus dependent. Specifically, we observed orientation and direction columns formed by sSC neurons retinotopically mapped to the edge of grating stimuli. For both excitatory and inhibitory neurons in sSC, orientation selectivity can be induced by the edge with their preferred orientation perpendicular to the edge orientation. Furthermore, we found that this edge-induced orientation selectivity is associated with saliency encoding. These findings indicate that the tuning properties of sSC neurons are not fixed by circuit architecture but rather dependent on the spatio-temporal properties of the stimulus.

A fundamental feature of the visual pathway in the brain, orientation selectivity plays a prominent role in visual processing and perception. Since the discovery of orientation-selective (OS) neurons in the cat primary visual cortex (V1)[1], the underlying mechanisms of orientation selectivity have been intensively investigated as a model system for understanding neural circuit computation[2–4]. In V1 of primates and cats, OS cells organize into highly structured orientation columns, where cells with the same orientation preference cluster together and orientation is smoothly mapped in space such that nearby cells have similar orientation[1,5–8]. In contrast, the spatial distribution of OS neurons in rodent V1 was found to follow a "salt-and-pepper" pattern, with neurons preferring different orientations intermingled in a random fashion[8–10]. The evolutionary, developmental, and circuit mechanisms underlying orientation columns remain hotly debated topics[11–16].

Outside the visual cortices[17–20], OS neurons have also been observed along the early visual pathways including retina[21–26] and dorsal lateral geniculate nucleus[20,26–29]. Recently, orientation selective neurons were also reported in mouse superior colliculus (SC)[30–33], a midbrain structure that integrates and transforms multisensory inputs into motor output such as saccades, head and body orienting, and aversive behaviors like escape and freezing[34–37]. In mouse, the superficial SC (sSC, stratum griseum superficiale and stratum opticum) receives input from >85% of retinal ganglion cells and visual cortices in a topographically organized and retinotopically aligned manner[38–41]. Many neurons within the sSC respond to visual stimulation and are selective for orientation or direction of motion[33,42,43].

Recently, several studies have reported a columnar organization for OS as well as direction-selective (DS) neurons in the mouse SC[30–32,44,45]. Even though the spatial patterns of these columns were sometimes inconsistent, neurons preferring gratings of the same orientation were found to form columns that were perpendicular to the SC surface throughout the retinorecipient layers. The existence of orientation columns in the mouse SC, in contrast to their absence in mouse V1, raises intriguing questions about the mechanisms and functions of orientation columns in the visual system, and provides a new model system in which these topics can be explored. However, these columns were not observed in other studies[46–48].

[1]Janelia Research Campus, Howard Hughes Medical Institute, Ashburn, VA 20148, USA. [2]Department of Molecular and Cell Biology, University of California, Berkeley, CA 94720, USA. [3]Department of Physics, University of California, Berkeley, CA 94720, USA. [4]Helen Wills Neuroscience Institute, University of California, Berkeley, CA 94720, USA. [5]Molecular Biophysics and Integrated Bioimaging Division, Lawrence Berkeley National Laboratory, Berkeley, CA 94720, USA. [6]Present address: Department of Diagnostic Radiology and Nuclear Medicine, University of Maryland School of Medicine, Baltimore, MD 21201, USA. [7]These authors contributed equally: Yajie Liang, Rongwen Lu. ✉e-mail: jina@berkeley.edu

By carrying out in vivo two-photon imaging over large areas of sSC in head-fixed awake mice, we discovered that neurons that retinotopically mapped to the edge of grating stimuli formed orientation as well as direction columns. We observed a smooth mapping of orientation tuning along the edges of circular gratings and a preference towards orthogonal orientations along the edges of square gratings. Moreover, the orientation preference of individual excitatory or inhibitory sSC neurons changed with a change in the orientation of the edge. Further experiments indicated that this edge-induced orientation selectivity is associated with saliency encoding. Together, our results suggest a novel mechanism that generates a columnar organization of stimulus-dependent orientation selectivity in the early processing of visual information.

## Results

### In vivo calcium imaging of sSC reveals orientation-selective responses at the edge of circular grating stimuli

We used a two-photon fluorescence microscope[49] and calcium imaging to characterize visually evoked responses over large areas of sSC in head-fixed awake mice (see Supplementary Table 1 and Methods for detailed experimental procedures and data analysis), with neurons expressing the genetically encoded calcium indicator GCaMP6s via viral transduction of AAV2/1.syn.GCaMP6s.WPRE.SV40. After removing the overlaying cortex and transverse sinus, we placed a cranial window above SC (Fig. 1a), which granted us chronic imaging access to a ~2-mm-diameter area of the dorsal SC (Fig. 1b, in vivo brightfield and widefield fluorescence images). For each mouse, we used a low-magnification microscope objective (4×, NA 0.28) to acquire the functional retinotopy map of sSC responding to an 80° × 80° visual

field of the right eye (Methods, Fig. 1c). The resulting map revealed a retinotopic relation between nasal visual field and anterior sSC, as well as between upper visual field and medial sSC, consistent with previous studies[30,42,50,51].

We then presented to the animal circular gratings drifting in 12 directions spanning 32° view angle (spatial frequency: 0.04 cycles per degree, cpd; temporal frequency: 1.5 Hz) and measured the calcium responses from sSC using the 4× objective (imaging depth centered at 80 μm; Fig. 1d). We found that pixels retinotopically mapped to the edge of the circular gratings had the largest calcium response. Unexpectedly, upon analyzing the orientation selectivity of the calcium response for each pixel, we found a ring of OS pixels in the same edge-corresponding area, with the preferred orientation smoothly varying along the ring and the opposite sides of the ring selecting for the same orientation.

Selecting regions of interest (ROIs) within the ring (ROIs 1–10, 56 μm in diameter, Fig. 1d) and plotting their calcium transients ΔF/F across all 12 drifting directions (Fig. 1e), we observed visually evoked activity with orientation selectivity, with ROIs on the opposite sides of the ring sharing the same preferred orientation, consistent with the pixelwise results. In contrast, example ROIs close to the center of the circular gratings had visually evoked activity of smaller calcium transients and reduced orientation selectivity (e.g., ROI 11, Fig. 1d, e); a ROI outside the grating retinotopic area had no visually evoked activity (ROI 12, Fig. 1d, e). These edge-induced effects in sSC were very robust. In all animals imaged with similarly sized circular grating stimuli (circular gratings of 31° or 34° diameters, see Supplementary Fig. 1 for data on 6 additional mice), we observed the same activity and OS patterns.

If these OS responses were evoked by the edge of the drifting gratings, changing the size and/or shape of the gratings should change

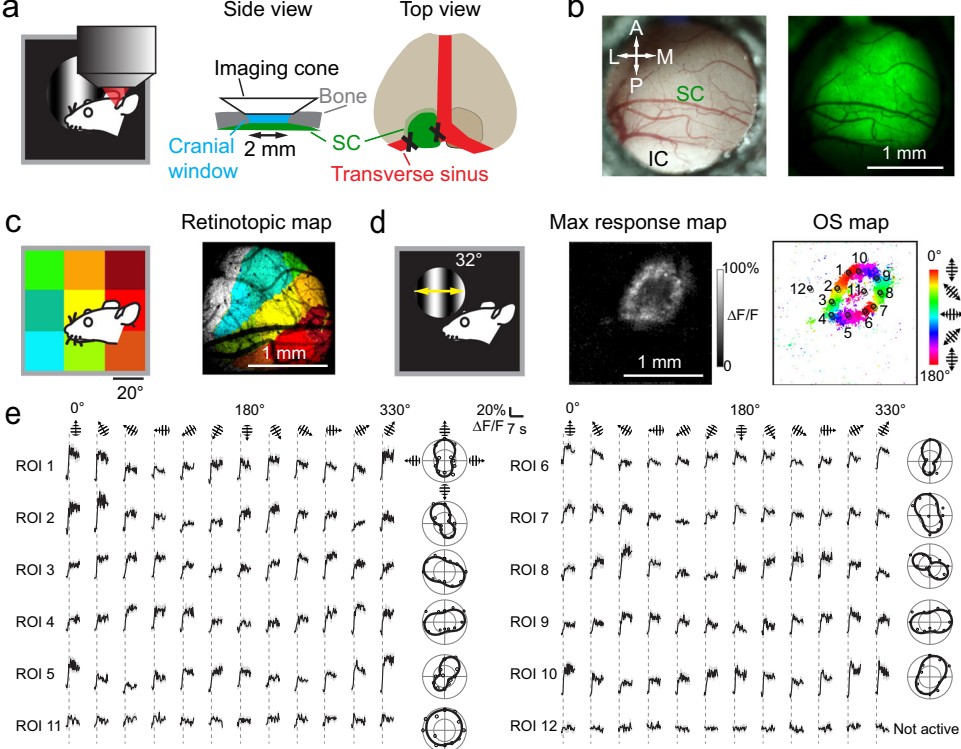

**Fig. 1 | In vivo calcium imaging of SC reveals orientation-selective responses at the edge of grating stimuli. a** Schematics of in vivo imaging and surgical preparation with cortex and transverse sinus overlying left SC removed and SC neurons transfected with AAV2/1.syn.GCaMP6s. **b** Brightfield and widefield fluorescence images of SC through a 2-mm-diameter cranial window. IC inferior colliculus. Representative results reproduced in *N* = 6 mice. **c** Example retinotopic map overlaying two-photon fluorescence image of left SC. Color denotes stimulus

position. **d** Circular gratings spanning 32° visual field and drifting in 12 directions, map of the maximal calcium signal evoked by these gratings, and map of orientation selective pixels (pixel size: 7 μm; color-coded by preferred orientation). **e** Example calcium transients (ΔF/F; ten-trial average, black trace; s.e.m., gray shade) and their normalized polar plots for ROIs in (**d**) (ROI diameter: 56 μm). Dashed line: transition from gray screen to drifting grating.

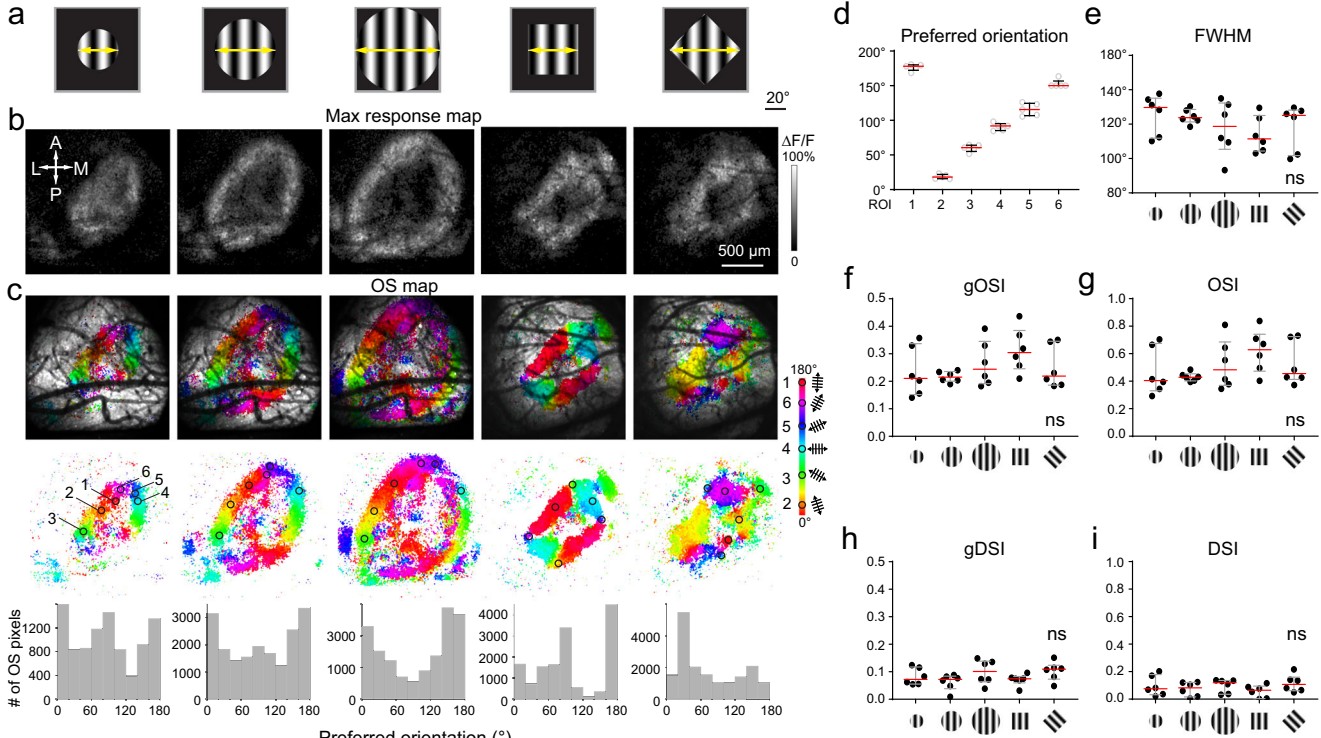

**Fig. 2 | Orientation-selective responses are specific to the size and shape of the drifting grating stimuli. a** Circular and square drifting grating stimuli, **b** their evoked maps of maximal calcium response, **c** maps of OS pixels (pixel size: 7 μm; color-coded by preferred orientation) and histograms of preferred orientation for all OS pixels in the imaging field of view. **d** Distributions of preferred orientations for the ROIs 1–6 from the 5 OS maps in (**c**). Orientation selectivity parameters, **e** FWHM, **f, g** gOSI and OSI, **h, i** gDSI and DSI for the 30 OS ROIs in (**c**) across the five different stimuli. ns no significant difference in parameter distributions across the five stimuli; $p > 0.05$, Kruskal–Wallis test. Error bars: 25 and 75 percentiles; Red lines: median. Source data are provided as a Source Data file.

the locations of the OS pixels. Indeed, when we presented circular drifting gratings of 38°, 58°, and 80° view angles, respectively, we observed rings of maximal response and orientation selectivity of increasing size in sSC (Fig. 2a–c, Columns 1–3), with similar preferred orientation distributions along the rings. When we presented a square drifting grating (Fig. 2a–c, Column 4), the maximal response and OS pixel maps adopted square shapes, produced by the SC neurons retinotopically mapped to the edge of the square gratings. The preferred orientations of OS pixels evoked by the square gratings were dominated by two orthogonal orientations. When the square gratings were rotated by 45°, the dominant orientations also shifted by ~45° (Fig. 2a–c, Column 5).

For each grating stimulus type in Fig. 2, we selected six ROIs of OS pixels with similar preferred orientations (ROIs 1–6, 56 μm in diameter, Fig. 2c; Preferred orientations: 176.5 ± 4.8°, 18.6 ± 3.3°, 59.8 ± 5.2°, 90.6 ± 5.4°, 115.6 ± 9.1°, 152.7 ± 6.0°, mean ± s.d., Fig. 2d). We then plotted their calcium transients and calculated their tuning curves (Supplementary Fig. 2), from which we obtained their orientation tuning parameters including full width at half maxima (FWHM, Fig. 2e), global orientation selectivity index (gOSI, Fig. 2f) and orientation selectivity index (OSI, Fig. 2g), as well as global direction selectivity index (gDSI, Fig. 2h) and direction selectivity index (DSI, Fig. 2i). No statistically significant difference was observed from the tuning parameter distributions across different grating patterns ($p > 0.05$, Kruskal–Wallis test).

Similar edge-induced effects were observed for drifting gratings of broad ranges of spatial and temporal frequencies (0.02–0.32 cpd, 0.5–16 Hz, Supplementary Fig. 3). Together, these experiments indicate that the observed edge-induced orientation selectivity was independent of grating size or shape and persisted over wide ranges of spatiotemporal frequencies.

## Edge-induced orientation selectivity also exists in mice with intact overlaying cortex and leads to orientation and direction columns in sSC

We tested whether the edge-induced effects on calcium response and orientation selectivity could be found in sSC with the overlaying cortex intact. Following a previously reported surgical approach[32], we carried out in vivo calcium imaging experiments on sSC neurons without removing cortex (Fig. 3, Supplementary Fig. 4). We pushed the transverse sinus anteriorly to expose the medial posterior sSC and virally transduced the expression of nuclear-targeted GCaMP6s in sSC neurons with AAV2/1.syn.H2B-GCaMP6s.WPRE.SV40 (Supplementary Fig. 5a). Imaging these neurons with a 10 × 0.6-NA water-dipping objective enabled us to study the tuning properties of individual sSC neurons over large imaging fields of view (Methods, Fig. 3a).

Retinotopically, medial posterior sSC corresponds to upper posterior visual field, which was at the edge of our 80°circular gratings. Consistent with the results obtained from cortex-removed SC, we found OS responses in sSC neurons retinotopically mapped to the circular grating edge, with their preferred orientations smoothly varying across the imaging fields (Fig. 3b). We quantified how much the preferred orientation differed with the inter-neuron distance for pairs of OS neurons within the imaging plane (Fig. 3c), which showed that neurons closer to one another had more similar orientation preference. Comparing the preferred orientation angle difference of neuron pairs that were nearest neighbors with those of randomly selected neuron pairs, we found that the nearest neighbors had significantly more similar orientations (two-sample Kolmogorov–Smirnov test). Across the depth of sSC, OS neurons had similar preferred orientations and formed orientation columns.

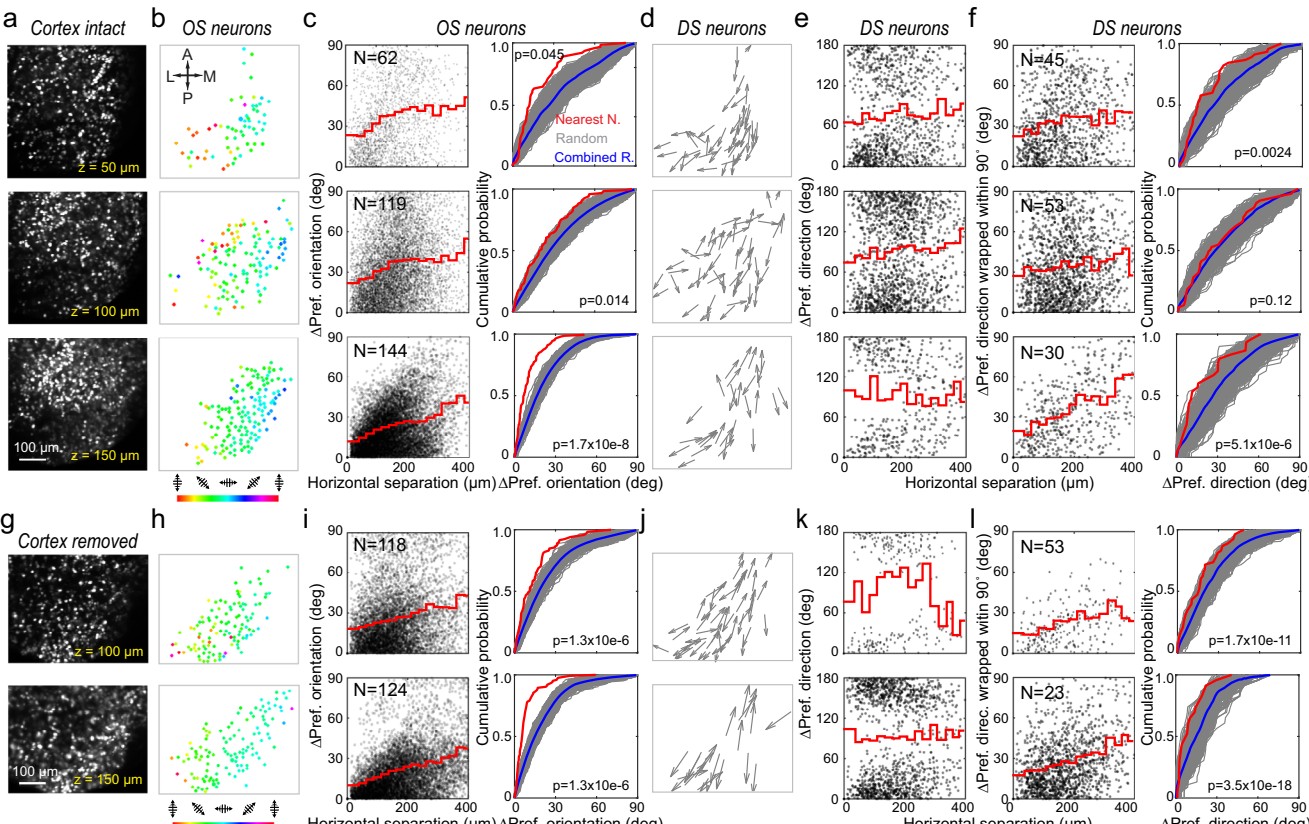

**Fig. 3 | Stimulus edge leads to orientation and direction columns in both cortex-intact and cortex-removed mice. a** sSC neurons expressing nuclear-targeted GCaMP6s at different depths in a cortex-intact mouse. **b** OS neurons mapped to the edge of 80° circular drifting gratings, color-coded by their preferred orientation. **c** Left: Horizontal separation for pairs of OS neurons versus difference in their preferred orientations (red curve: averages of 25-μm-wide bins). Right: Cumulative distributions of preferred orientation difference between nearest neighbors (red curve), randomly selected neuron pairs (1000 repeats, gray curves), and the average of random distributions (blue curve). *p* value: two-sample Kolmogorov–Smirnov test between red and blue curves. **d** DS neurons with their preferred directions of motion indicated by arrows. **e** Horizontal separation for pairs of DS neurons versus difference in their preferred directions. **f** Left: Horizontal separation for pairs of DS neurons versus difference in their preferred directions wrapped to (0, 90°); Right: same analysis as in (**c**). Representative result from five mice. **g–l** Same as (**a**, **b**, **c**, **d**, **e**, **f**), respectively, but acquired from a mouse with overlaying cortex removed. Representative result from three mice. Statistical tests were two-sided.

We further investigated the direction-selective (DS) population, which we defined as OS neurons with DSI larger than 0.5 (e.g., at least 3× larger response to the preferred direction than the opposite direction). In contrast to the OS neurons, DS neurons within the same fields-of-view (FOVs) do not exhibit straightforward clustering of their preferred directions (Fig. 3d). Instead, nearby DS neurons tend to prefer similar or opposite directions of the same motion axis (Fig. 3e). Wrapping the preferred direction difference to (0, 90°) (i.e., if direction difference is larger than 90°, use its supplementary angle), we found this clustering of similar/opposite direction preference of nearby neurons to be statistically significant (Fig. 3f; two-sample Kolmogorov–Smirnov test). The clustering of the wrapped direction preference was observed across multiple depths, forming columns.

The same trends, including the formation of orientation and direction columns, were also observed in mice with the cortex overlaying the sSC removed (Fig. 3g–l). Furthermore, analysis on FOVs retinotopically mapped away from the stimulus edges indicated a lack of orientation clustering (Supplementary Fig. 4). Together, these results indicate that the presence of a stimulus edge induces orientation selectivity in individual sSC neurons in both mice with intact cortex and mice with overlaying cortex removed, and leads to orientation and direction columns in mouse sSC.

Given that the presence of these columns was found to be correlated with increased activity (Figs. 1, 2, Supplementary Figs. 1, 3), in separate experiments, we investigated whether sSC neurons at the edge of the stimuli in intact brains also had increased activity. We presented 80° circular drifting gratings to head-fixed mice with intact skull for two hours, then perfused and sectioned their brains for immunostaining against c-Fos, a marker of neuronal activity[52]. Consistent with in vivo imaging results, neurons with heightened c-Fos expression were found to distribute throughout sSC (Supplementary Fig. 5b). Reconstructing the brain volume from coronal sections with c-Fos labeling and viewing it dorsally, we found a ring-like pattern formed by the most active sSC neurons, presumably evoked by the circular edge of the 80° gratings.

## sSC neurons retinotopically mapped to edges of grating stimuli change their preferred orientation in response to changes in edge orientation

We next investigated the orientation selectivity of individual sSC neurons in response to changes in grating edge orientation. We used a 16 × 0.8-NA objective for two-photon fluorescence imaging of neurons expressing cytosolic GCaMP6s in sSC exposed by cortical removal. We used adaptive optics (AO) to correct the system and cranial window aberrations to minimize neuropil contamination[49,53].

We first presented the mouse with 80° circular drifting gratings and imaged the sSC neurons retinotopically mapped to the center of the gratings up to 180 μm depth (Fig. 4a, Supplementary Fig. 6, 10 imaging sessions, 6 mice). For neurons exhibiting visually evoked activity, we found 80% to be OS and characterized their orientation

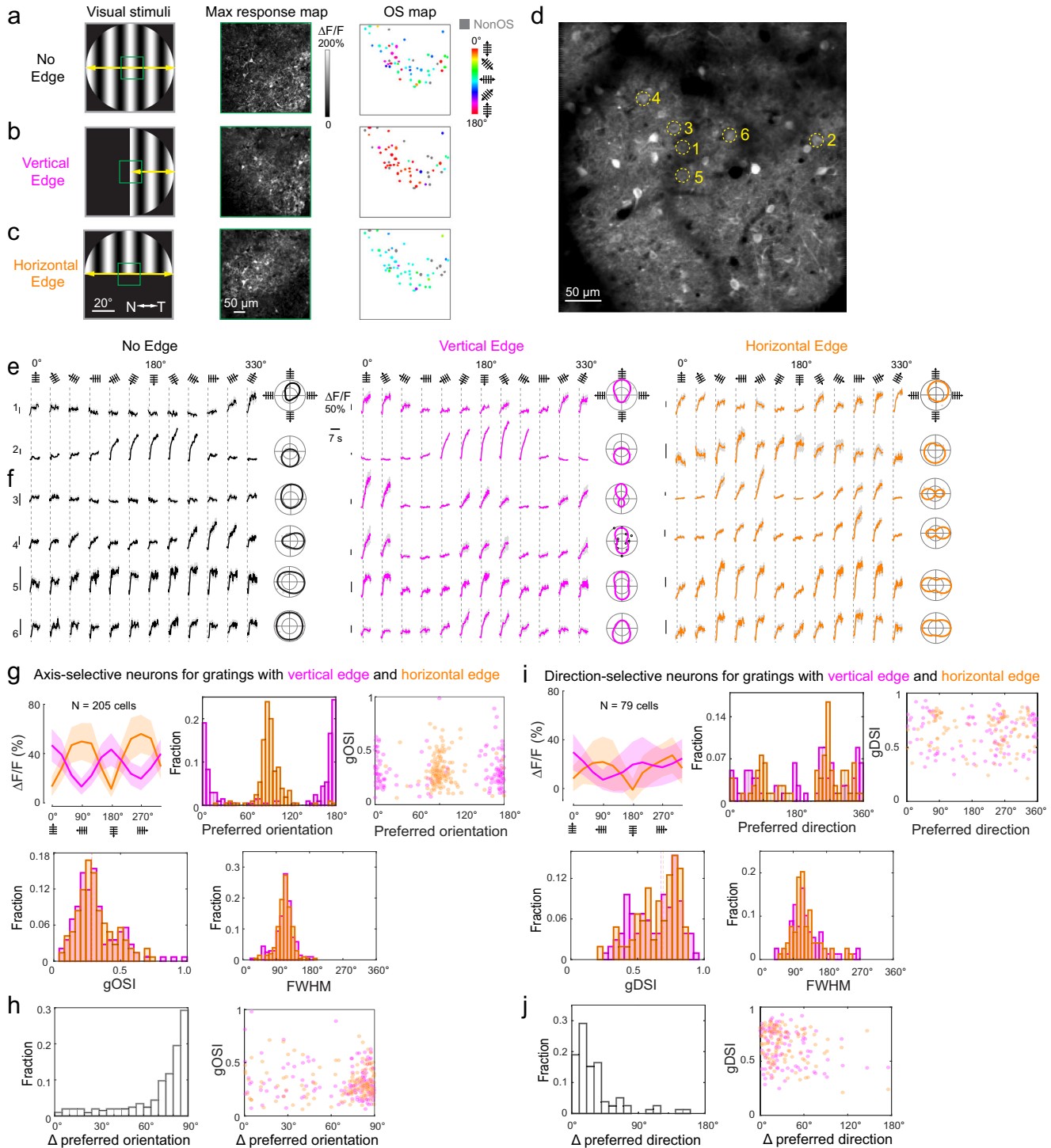

**Fig. 4 | sSC neurons change their orientation selectivity in response to the presence of grating edges of distinct orientations.** Maximal response and OS maps for example SC neurons whose receptive field mapped to screen center, where drifting gratings (**a**) without a central edge, (**b**) with a vertical edge, or (**c**) with a horizontal edge were presented. N and T nasal and temporal visual field directions. **d** Zoomed-in view of neurons either (**e**) maintaining or (**f**) switching their preferred orientation in response to grating edges through their receptive fields. Representative results reproduced in *N* = 6 mice. For axis-selective neurons in response to gratings with vertical or horizontal edge: **g** (Left to right, top to bottom) average calcium response ΔF/F, preferred orientation distribution, preferred orientation versus gOSI, and their OSI and FWHM distributions; **h** Change in preferred orientation for individual neurons and change in preferred orientation versus gOSI; 205 neurons from six mice. For direction-selective neurons in response to gratings with vertical or horizontal edge: **i** (Left to right, top to bottom) average calcium response ΔF/F, preferred direction distribution, preferred direction versus gDSI, and their DSI and FWHM distributions; **j** Change in preferred direction for individual neurons and change in preferred direction versus gDSI; 79 neurons from six mice. Shaded error bands: **e**, **f** s.e.m.; **g**, **i** s.d.

selectivity properties including their orientation selectivity index (OSI), global OSI, direction selectivity index (DSI), global DSI, and full width at half maximum (FWHM) (Methods). We then classified these OS neurons as axis selective (AS; DSI < 0.5; 49% of all neurons) or direction selective (DS; DSI ≥ 0.5; 31% of all neurons). Our results (Supplementary Fig. 6), including the observation of the fraction of DS neurons decreasing with depth, are consistent with previous reports[46,54].

We then recorded calcium responses in the same neurons while blocking either the left or the bottom half of the gratings, which generated a vertical (Fig. 4b) or horizontal edge (Fig. 4c) through the retinotopic fields of these neurons (Fig. 4d). The majority of neurons that were OS with full grating stimuli remained orientation selective (93% and 97% for vertical and horizontal edge, respectively). Comparing the calcium responses and tuning curves of the same neurons under these three conditions, we found that, whereas some AS and DS neurons maintained their orientation and direction preferences (Fig. 4e), more neurons changed their preferred orientation in response to changes in orientation of stimulus edge (Fig. 4f), consistent with the larger FOV pixel-based analysis above.

Characterizing individual neuron's tuning properties, we found that, for neurons that were AS under both vertical- and horizontal-edge grating stimuli, their preferred orientations were highly dependent on the edge orientation (Fig. 4g, h). With vertical-edge gratings, AS neurons predominantly selected for horizontally oriented gratings. These same neurons, when stimulated by horizontal-edge gratings, changed their preferred orientation by ~90° and became selective for vertically oriented gratings, with similar distributions for gOSI and FWHM of their tuning curves. In contrast, neurons that were direction-selective

under these two types of grating stimuli were less likely than the axis-selective neuron population to have large changes in their direction preference (Fig. 4i, j). Vertical and horizontal edges instead activated distinct subsets of DS neurons preferring different motion directions. These properties appeared to be independent of how orientation-selective (for AS neurons) or direction-selective (for DS neurons) these neurons were, as neurons with large or small changes in their preferred orientation/direction had their gOSI/gDSI values within similar ranges (Fig. 4h, j). The same trends were observed for interneurons labeled by injecting AAV2/1.flex.syn.GCaMP6s into the sSC of Gad2-IRES-Cre mice (Supplementary Fig. 7).

### Edge-induced changes in orientation selectivity of sSC neurons are distinct from those induced by moving texture stimuli

In addition to drifting gratings moving in directions orthogonal to grating orientation, we also measured the responses of sSC neurons towards three sets of texture stimuli composed of oriented moving bars. These experiments were inspired by previous work in the ferret V1, where changing the axis of motion of such stimuli resulted in striking shifts in the population orientation-tuning of V1 neurons[55]. We asked whether the same population tuning shifts happen in sSC, and if they do, how the population tuning shifts manifest themselves in tuning of individual sSC neurons.

We imaged at single-cell resolution the responses of neurons expressing cytosolic GCaMP6s in sSC exposed by cortical removal towards four sets of stimuli extending over 30° visual field (10 × 0.6-NA or 16 × 0.8-NA objective, 7 FOVs in 4 mice, $N$ = 2204 visually responsive neurons; Fig. 5). They included gratings moving along a direction orthogonal to the grating orientation ("grating"; 0.04 cycles/degree,

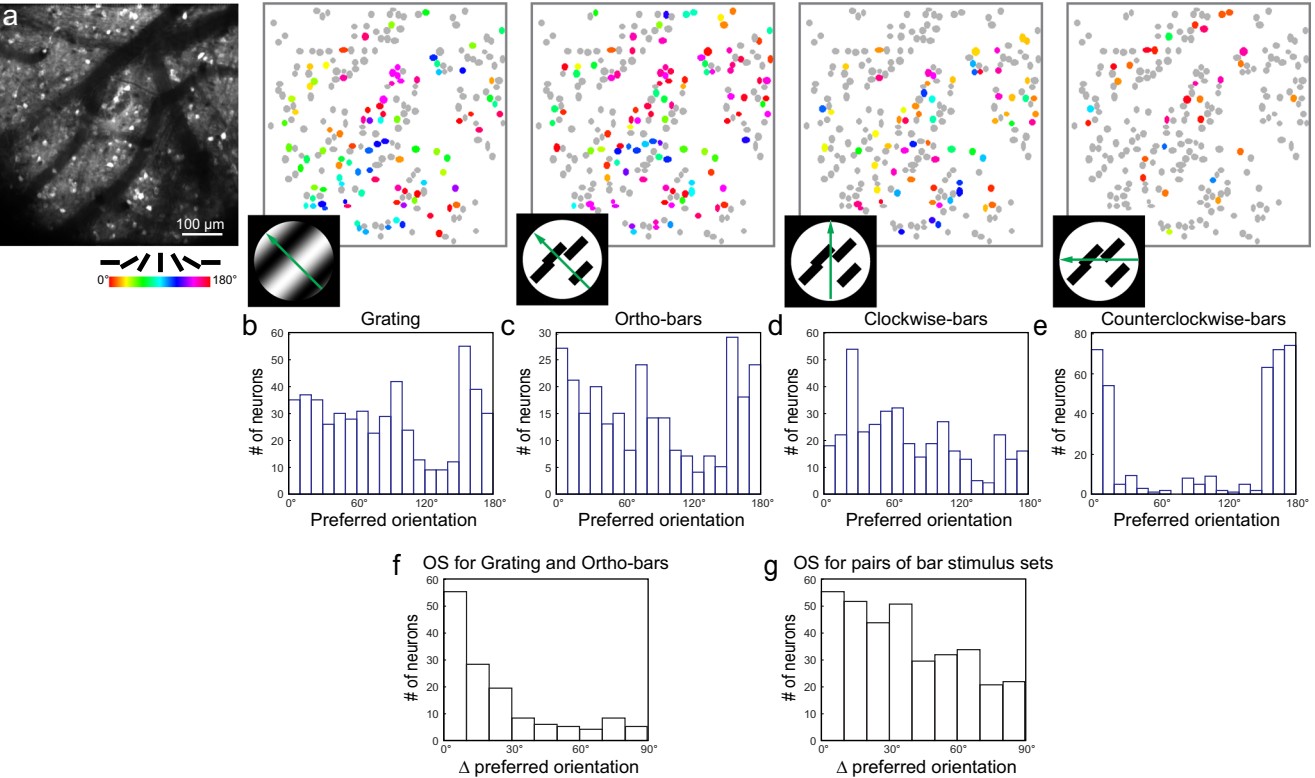

**Fig. 5 | Orientation tuning of sSC neurons in response to moving texture stimuli. a** A representative FOV of sSC neurons with the OS neurons color-coded by their preferred grating/bar orientation for grating/moving bar stimulus sets. Representative results reproduced in $N$ = 4 mice. **b–e** Histogram distributions of preferred orientation angles for neurons that are orientation selective towards **b** grating, **c** ortho-bars, **d** clockwise-bars, and **e** counterclockwise-bars stimulus

sets, respectively. **f** Histogram distribution of difference in preferred orientation angles for neurons that were OS for both grating and ortho-bars stimulus sets. **g** Histogram distribution of difference in preferred orientation angles for neurons that were OS for ortho-bars and clockwise-bars, ortho-bars and counterclockwise, or clockwise-bars and counterclockwise stimulus sets. Source data are provided as a Source Data file.

1.5 Hz), bars moving in a direction orthogonal to the orientation of the bars ("ortho-bars"; 10° in length, same speed as grating), and bars moving obliquely with the bar orientation being 45° clockwise ("clockwise-bars") or 45° counter-clockwise ("counterclockwise-bars") from its motion direction. From the calcium responses of individual neurons, we determined the preferred orientations of individual sSC neurons (Fig. 5a).

We first investigated the population orientation tuning elicited by these stimulus sets. We found that grating and ortho-bars stimuli led to similar population tuning as indicated by the histogram distributions of preferred orientations (Fig. 5b, c). But when we changed the axis of motion of the bar stimuli from orthogonal to oblique relative to the bar orientation, we observed shifts in population tuning (Fig. 5d, e). These observations are consistent with previous work[55] in V1 and may be similarly explained by the stimulus spatiotemporal energy model[56–58].

We then evaluated how tuning changes for individual neurons, by identifying neurons that were OS in pairs of stimulus sets and investigating how they changed their preferred orientation angles. For individual neurons that were OS for both grating and ortho-bars, most did not change their preferred orientation (Fig. 5f). For neurons that were OS for two bar stimulus sets, a higher fraction of neurons exhibited a substantial change in their preferred orientation angle (Fig. 5g). However, because only a minority of OS neurons tuned in one bar stimulus set were also OS in another bar stimulus set (0.33 ± 0.09, mean ± s.d.), the fraction of OS neurons for a specific stimulus set that changed their preferred orientation in response to changes in motion of these texture stimuli was low. In other words, the observed population tuning changes were mostly generated by different populations of neurons that were tuned for different stimulus sets. This is distinctively different from the orientation preference changes induced by grating edges described in the previous section, where a majority of OS neurons changed their orientation preference in response to changes in edge orientation (Fig. 4h).

### Edge-induced orientation selectivity in sSC is associated with saliency encoding

On the population level, for sSC neurons with visually evoked activity across all three stimulus conditions (363 neurons from 6 mice), the presence of an edge in grating stimuli significantly increased response magnitude and orientation selectivity of neurons retinotopically mapped to the edge (Supplementary Fig. 8). Given the prominent role that the SC plays in bottom-up saliency[59–61], we further investigated whether the edge-induced orientation selectivity in sSC is related to saliency encoding. To encode saliency in visual space, the neural response to visual locations with higher saliency should be elevated. The edge of drifting gratings represents an abrupt transition between the presence and absence of a strong visual stimulus, and thus possesses high saliency. In both cortex-intact and cortex-removed mice, our results indicated that sSC activity patterns indeed correlated with saliency in the visual space, with borders of the drifting grating stimuli evoking the strongest response in aligned sSC neurons (Figs. 1–3, Supplementary Fig. 5b).

To further evaluate saliency encoding, we measured how sSC neurons responded to visual stimuli with varied saliency. We compared the responses to concentric grating stimuli (29° and 58° for inner and outer diameters) with the inner grating moving in the same ("iso"), orthogonal, or opposite direction to that of the outer grating. All concentric stimuli evoked orientation-selective responses from sSC neurons (Fig. 6a, b). For cells retinotopically matched to the edge of the inner grating (identified as the OS pixels when only inner gratings were presented; donut-shaped ROIs encircled by dashed lines in Fig. 6b), maps of their maximal calcium responses indicated increasingly strong responses with the increase of stimulus saliency from iso, orthogonal, to opposite gratings, consistent with the prediction for activity strength in saliency encoding. The same trend was observed

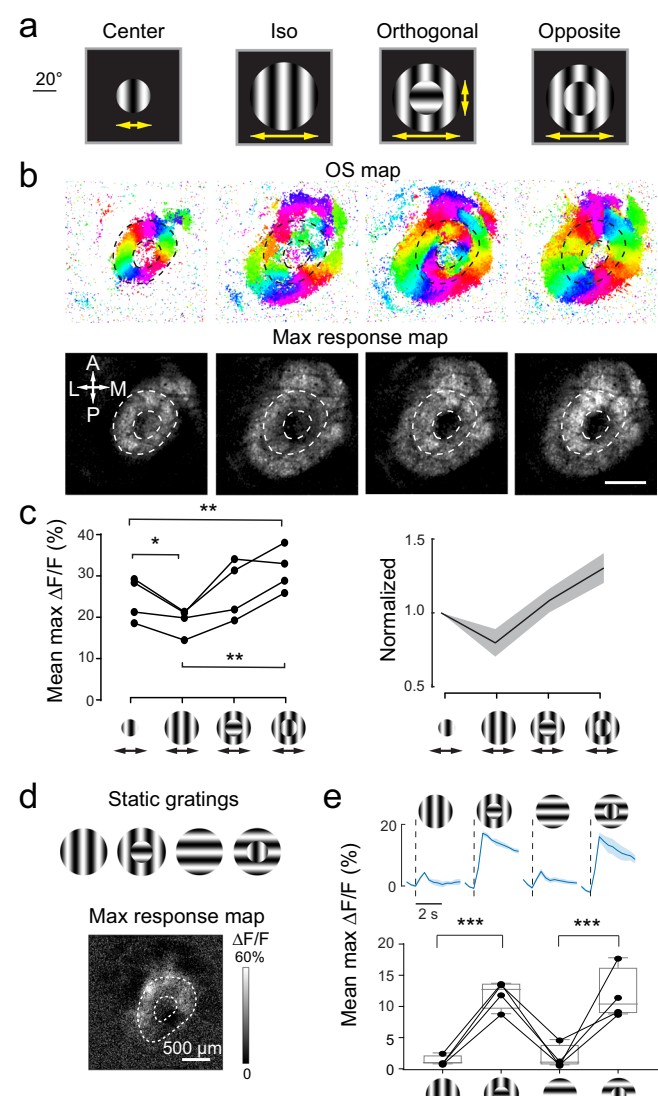

**Fig. 6 | Edge-induced orientation selectivity is associated with saliency encoding in the sSC. a** Drifting grating stimuli and **b** OS and maximal response maps of sSC neurons. **c** Averaged maximal responses of pixels retinotopically mapped to the inner grating edges (area between dashed circles) from 4 mice. Shaded regions: s.d. **d** Static gratings and their evoked maximal response map (to gratings with orthogonal centers). **e** (Upper) temporal traces of the averaged maximal calcium transients from one example mouse and (Lower) box-whisker plots of the averaged maximal responses between the dashed circles from 4 mice. Box represents median and 25th–75th percentiles; Whiskers in Tukey style (plus or minus 1.5 times IQR). One way ANOVA test with post-hoc Tukey's multiple comparison test was used for data in (**c**, **e**): *$p < 0.05$, **$p < 0.01$, ***$p < 0.001$. Shaded error bands in (**c**, **e**): s.d. In (**c**), $p = 0.041$ for comparison between 1st and 2nd groups from left; $p = 0.0040$ for 1st vs 4th groups; $p = 0.0047$ for 2nd vs 4th groups. In (**e**), $p = 0.00011$ for the comparison between 1st and 2nd groups and $p = 0.0047$ for 3rd vs 4th group. Source data are provided as a Source Data file.

for the averaged maximal responses, which were significantly higher for more salient stimuli (Fig. 6c).

Another key feature of a saliency map is that it should be agnostic to detailed visual feature characteristics, which predicts that its representation in sSC neuronal activity should not be limited to moving stimuli. To test this prediction, we presented the mice with stationary gratings that were either uniformly horizontal/vertical or with an inner area that was orthogonal to the outer grating (Fig. 6d). As expected, we observed much larger responses from sSC neurons at the

edge between orthogonal inner and outer gratings (Fig. 6e). Together, these results suggest that edge-induced effects were associated with saliency encoding and that sSC produces a saliency map[62].

## Discussion

In both cortex-removed and cortex-intact preparations, we found that edge-induced OS responses spanned the entire retinorecipient sSC, with neurons preferring the same orientation forming vertical columns. However, instead of having preferred orientations that are intrinsic, stereotypical, and largely stimulus-independent as in V1 of primate and cat[63], sSC neurons within these edge-induced orientation columns have their orientation preference dynamically modified and contextually modulated by the stimuli. Similarly, the stimulus edges also induce direction columns. These findings are consistent with a recent work[48], which used an immersive dome display to cover almost the entire visual field of the mouse and reported a lack of direction columns in the mouse sSC, likely because there was no stimulus edge within the retinotopic field of the investigated neurons. In contrast, prior studies that observed orientation and direction columns in sSC utilized smaller screen sizes. If these columns were caused by edges of the area within which visual stimuli were presented, the inconsistencies among these studies on the structures of the orientation and direction columns can be explained by the differences in size and location of visual stimulation area.

Compared to mouse V1, neurons in mouse SC exhibit less activity modulation by behavioral states as well as lower response variability[45,47]. These observations are consistent with the notion that sSC is involved in bottom-up salience computation, with its activity pattern determined more by stimulus saliency than by behavioral states or top-down signals. By imaging the activity of neurons over large areas of the mouse sSC, our results provide direct evidence of mapping between activity patterns of sSC neurons and the saliency map of the visual space, with heightened activity observed at the salient boundaries between distinct visual stimuli. Whereas this mapping was long proposed theoretically[59,64,65], this is the first time, to our knowledge, such a map has been observed experimentally.

Together, our data reveal a novel mechanism for the generation of columnal OS responses in the sSC, one that dynamically depends on the specifics of visual stimulation and thus is distinct from those underlying the orientation columns observed in primate and cat V1. Furthermore, we have discovered that these OS responses are associated with saliency encoding by sSC.

Our data on the population activity of the sSC neurons can be understood within the framework[56–58,66] of the spatiotemporal energy model (SEM). In SEM, the spatiotemporal receptive field of a neuron in space-time $(x,y,t)$ can also be described in the spatial and temporal frequency domain $(w_x,w_y,w_t)$. To determine how neurons respond to a specific visual stimulus, one looks at the spatiotemporal frequency domain representation of the stimulus. Neurons with receptive fields inside the volume occupied by the visual stimulus in the spatiotemporal frequency domain are activated by the stimulus. Therefore, SEM predicts that a stimulus with a larger volume in the spatiotemporal frequency space elicits responses from a larger population of neurons. This is consistent with our maximal response map analysis (Figs. 1d, 2b, 6b; Supplementary Fig. 1a), where the strongest population responses were located at the edge of our stimuli. Discontinuities in $(x,y)$ at the grating edge lead to a larger volume in the frequency domain and thus evoke responses from more neurons. SEM can also explain our observation of heightened activity at the salient boundaries between distinct visual stimuli: The salient boundaries are discontinuous by definition, and thus activate more neurons. SEM is also successful in explaining the increase of response magnitude with the increase in the spatial frequency of grating stimuli (Supplementary Fig. 3b):

gratings with higher spatial frequency are composed of narrower line segments, the ends of which occupy a larger volume in $(w_x,w_y,w_t)$. These gratings thus elicit responses from more neurons and lead to a larger maximal response, as we observed experimentally.

However, questions remain. Does sSC generate the edge-dependent orientation selectivity and compute the saliency map locally, or does it inherit these characteristics from retina and/or visual cortices? Given that we observed these properties in sSCs of cortex-removed mice, it is unlikely that the sSC inherits its saliency responses from cortex. In in vivo calcium imaging experiments in L2/3 neurons of the mouse V1, we did not observe similar edge-induced effect (Supplementary Fig. 9). In this aspect, our results are in line with prior studies in both rodents and primates that removing V1 inputs has limited impact on the visual response properties of sSC neurons[30,33,67–69]. However, the possibility that saliency computation starts in retina remains. Given the existence of similar center-surround interactions and suppression by stimuli in the extra-classical surround in the retina[70,71], saliency mapping may already exist at the retinal output, which can be investigated by imaging the axonal inputs from the retina in sSC[72]. Knowing how saliency is encoded in the retinal inputs would then allow us to determine the contribution of circuit computation within sSC itself.

The saliency map formed in the sSC likely propagates through both cortical and subcortical pathways to influence sensory processing and behavior. Projections from sSC to the intermediate layer of SC may allow the integration of the visual map with other sensory modalities, forming a multimodal saliency map that in turn directs orienting and defensive behaviors through projections to motor areas[73–76]. Through SC projections to visual thalamus, it can also affect attention modulation and saliency processing in visual cortices[77–79] and contribute to form processes such as perceptual figure-ground segregation[67,80,81]. Utilizing behaving animals in future inquiries would help to elucidate the ethological roles that the stimulus-dependent orientation selectivity of sSC neurons may play.

## Methods

All experimental protocols were conducted according to the National Institutes of Health guidelines for animal research and were approved by the Institutional Animal Care and Use Committee at Janelia Research Campus, Howard Hughes Medical Institute and the University of California, Berkeley.

### Mice

Adult C57BL/6 mice (Jackson Laboratory) and transgenic mice of both sexes were used, including Gad2-IRES-cre (Jax no. 010802). They were housed under a reverse light cycle at ~70 °F and ambient humidity. Sample sizes (number of mice, cells and/or fields of view, FOVs) for each experiment are stated in main text. AAV viruses were obtained from Virus Services of Janelia Research Campus, HHMI.

### SC exposure, viral injection, and cranial window implantation

Mice were anaesthetized with isoflurane (1–2% by volume in $O_2$) and given the analgesic buprenorphine (SC, 0.3 mg per kg of body weight). They were placed in a stereotaxic device with eyes covered with ophthalmic ointment. A 3-mm-diameter craniotomy was performed over the left SC to carefully expose cortex and transverse sinus. The exposed brain was constantly irrigated with artificial cerebrospinal fluid to keep tissue moist. Two different preparations, cortex-removed and cortex-intact, respectively, were then made.

For cortex-removed preparation, we first sutured the left transverse sinus. An 8-0 suture was doubled and guided by a pair of fine forceps to go under the transvers sinus at its midpoint across the left SC. The double suture was then separated into two single sutures, and two knots about 0.5 mm away from each other were tied to stop blood

flow in the left transverse sinus, which was then severed between the two knots. The two ends of the severed transverse sinus were carefully peeled off from the underlying brain tissue by lifting the knots. This procedure completely removed left transverse sinus and exposed the underlying SC. The cortex overlying the anterior portion of the SC was then slowly aspirated as described in previous studies[30,32,50,51]. After the SC was exposed, virus injection was performed using a glass pipette (Drummond Scientific Company) beveled at 30° with a 15–20-μm-diameter opening and back-filled with mineral oil. A fitted plunger controlled by a hydraulic manipulator (Narashige, MO10) was inserted into the pipette and used to load and inject the viral solution. Virus-containing solution (AAV2/1.syn.GCaMP6s or AAV2/1.syn.H2B-GCaMP6s for wild-type mice, AAV2/1.syn.Flex.GCaMP6s for Gad2-IRES-Cre mice, $1 \times 10^{13}$ infectious units per ml) was injected 0.4 and 0.2 mm below the surface of SC (15 nL for each depth) over multiple sites spaced 0.3–0.4 mm apart across the exposed SC surface. After virus injection, a glass window made of a single 2-mm-diameter glass window (Fisher Scientific No. 1.5) glued to a 3D-printed plastic cone (2-mm and 4-mm-diameter opening on either side, 8-mm height) was embedded in the craniotomy and sealed in place with cyanoacrylate glue.

For cortex-intact preparation, a glass plug was first made by gluing two layers of triangular glass windows to a 2.5-mm-diameter round coverglass, which was then glued to a donut-shape glass using UV curable glue (Norland Optical Adhesive; all windows laser-cut from Fisher Scientific no. 1.5 coverslips). As described in a previous study[32], this glass plug was then used to push transverse sinus anteriorly to expose SC without cortex removal and then sealed in place with cyanoacrylate glue.

After window implantation, a titanium headpost was attached to the skull with cyanoacrylate glue and subsequently dental acrylic (OrthoJet). Animals were given Buprenorphine (0.1 mg/kg, i.p.) and Ketoprofen (0.1 mg/kg, i.p.) for 48 h post-operatively. A group of mice ($N = 4$) were only installed with headpost without craniotomy for visual stimulation and subsequent c-Fos immunostaining.

### Viral injection and cranial window installation in visual cortex
Craniotomy and imaging window implantation over the primary visual cortex were carried following a procedure similar to those for SC. A 2.5-mm-diameter craniotomy was performed over left V1 (center: 2.5 mm lateral from midline, 0.5 mm anterior to Lambda) with dura left intact. Virus injection of AAV2/1.syn. GCaMP6s was then performed as described above. A glass window made of a single No. 1.5 coverslip was embedded in the craniotomy and sealed with VetBond. A titanium head-post was attached to the skull with dental acrylic.

### Two-photon imaging
All imaging was performed on head-fixed awake mice after at least two weeks of recovery in single or paired housing. Before imaging and one week after surgery, mice were habituated to experimental handling and head fixation. During both habituation and imaging, each mouse had its body restrained under a half-cylindrical cover, which reduced struggling and prevented substantial body movements such as running[28]. Habituation was repeated 3–4 times for each animal, and each time for 15–60 min. Each imaging session lasted 45 min to 2 h, with data collected from multiple imaging planes within the same mouse. GCaMP6s and H2B-GCaMP6s fluorescence was excited at 940 nm (InSight Deepsee, Spectra-Physics) under a Olympus 4× 0.28-NA, a Olympus 10× 0.6-NA, or a Nikon 16× 0.8-NA objective using a homebuilt two-photon fluorescence microscope, with all images collected at 3 Hz[49].

### Visual stimulation: general setting
Visual stimuli were generated using Psychophysics Toolbox[82] and presented to the right eye of the mouse by back projection on a screen made of Teflon film using a custom-modified DLP projector[83]. The screen was positioned 15 cm from the right eye, covering 80° × 80° degrees of visual space and oriented at 55° to the long axis of the animal. The projector was modified to provide equilength and linear frames at 360 Hz (designed by A. Leonardo, Janelia Research Campus, and Lightspeed Design, model WXGA-360). Its lamp was replaced by a holder for a liquid light guide, through which visible light (450–495 nm) generated by a LED light source (SugarCUBE) was delivered to the screen[83]. The maximal luminance measured at the location of animal eyes was 437 nW/mm². During visual stimulation, the total luminance over the entire screen was kept constant.

### Visual stimulation: retinotopic mapping
The retinotopic map was measured by sweeping a drifting black bar through the gray screen with the same size as described above. Drifting bars were 3° in width and drifted at a speed of 11° per second, followed by 4 s of gray screen until the next bar appears with a different orientation and drifting direction. In each trial, one out of eight drifting directions (from 0° through 315° with a step of 45° in a pseudorandom sequence) was presented. Seven trials were repeated for each direction. For each trial, two-photon imaging of SC started at the appearance of each bar and lasted 9 s.

### Visual stimulation: orientation selectivity test
To determine orientation selectivity (OS), drifting circular or square sinusoidal gratings of 12 directions (0–330°, with 30° increments) and varied size were presented in a pseudorandom sequence. Each stimulus trial was composed of a 4-s gray screen, a grating stimulus lasting for 7 s, and then another 2-s gray screen. Gratings had 100% contrast, a spatial frequency of 0.04 cycles per degree, and a temporal frequency of 1.5 Hz. Each stimulus was repeated 10 times. Two-photon fluorescence signal was acquired for 1 s during the presentation of gray screen and for 7 s during the drifting grating presentation. For pixel-based analysis over large FOVs, images were collected using the 4× 0.28-NA over a 1.8 mm × 1.8 mm FOV at a pixel size of 7 μm. For experiment using the 10× objective, images were collected over a 0.84 mm × 0.84 mm area at a pixel size of 3.3 μm. For experiments using the 16× objective, images were collected over a 0.38 mm × 0.38 mm FOV at a pixel size of 1.5 μm.

Orientation selectivity tests were performed with gratings of different spatial or temporal frequencies. Spatial frequency of 0.01, 0.02 0.04, 0.08, 0.16, 0.32 cycles per degree (cpd) was presented with temporal frequency fixed at 1.5 Hz. Temporal frequency varied between 0.5 and 16 Hz (0.5, 1, 2, 4, 8, 16 Hz) when spatial frequency was fixed at 0.04 cpd.

### Visual stimulation: moving texture stimulation
Moving bars of 12 orientations (0–330°, with 30° increments) were presented in a pseudorandom sequence. Direction of movement was either orthogonal, rotated 45° clockwise, or rotated 45° counter-clockwise relative to the bars' orientation. Each stimulus trial was composed of a 6-s gray screen followed by bar stimuli lasting for 7 s. Bars were black on a white circular background with a diameter of 30° visual angle. Each bar was 10° long, 3.5° wide, and moved at a temporal frequency of 1.5 Hz. Each bar orientation was repeated either 6 or 10 times.

### Visual stimulation: stimuli with a concentric structure
For stimuli with a concentric structure, the inner circle extended over 29° visual field and the outer circle extended over 58° visual field. The orthogonal and opposite grating stimuli had 100% contrast. Duration of visual stimulation and timing of two-photon acquisition were the same as those described in the orientation selectivity tests.

## Visual stimulation: stationary gratings

For each stationary grating trial, mice were presented with a gray screen for 4 s, followed by one of the four static grating patterns for 6 s (full vertical grating, vertical grating with an inner circle of horizontal grating, full horizontal grating, or horizontal grating with an inner circle of vertical grating). Each stimulus trial was repeated 10 times. Gratings had 0.04 cycles per degree with 100% contrast. Two-photon fluorescence signal was acquired for 1 s during the gray screen and for 3 s during the static grating presentation.

## Visual stimulation: half-circular gratings with vertical or horizontal edges

For the change of orientation test at single-cell resolution, the illumination area on the screen that activates the SC area within the imaging FOV of the 16× objective was first identified using a 10° view angle drifting grating circle sweeping through the whole screen. Then the drifting grating was expanded to 80°, and two sets of visual patterns were presented, in which a rectangular opaque screen was used to cover either the nasal or the lower half of the illumination circle, creating a vertical or horizontal dark edge across the receptive field of the imaging area in the SC, respectively. The duration of visual stimulation and timing of two-photon imaging were the same as the orientation selectivity test.

## c-fos measurement and histology

For c-fos immunostaining, we installed head-posts to intact skull of mice. Two weeks later, mice were head-fixed and presented with drifting gratings (80° view angle) for two hours. Immediately after, mice were overdosed with isoflurane and perfused with PBS and then 4% paraformaldehyde (PFA). The brain was immersed in 4% PFA overnight. Coronal slices 50 μm thick were cut from the fixed brain embedded in 5% agar using a vibrating blade microtome (V1200S, Leica). For immunostaining, we applied primary antibody (anti-c fos, Cell Signaling, 1:500) and subsequent secondary antibody (Alexa Fluro 594-conjugated goat anti-rabbit, ThermoFisher, 1:500) to free floating brain sections including the SC. All brain slices were then mounted in Vector Shield mounting solution. Images were captured using a stereomicroscope at low zoom (2–4×) or at high zoom with Zeiss Apo-Tome 2.0 (20×/0.8NA). Images were then processed with ImageJ[84] and Vaa3D was used for 3D reconstruction[85].

## Analysis of two-photon imaging data

Imaging data were processed with ImageJ and with custom programs written in Matlab (Mathworks). To correct brain motion during imaging, images were registered with an iterative cross-correlation-based rigid registration algorithm[28].

## Retinotopic mapping

Images were first smoothed spatially with a Gaussian filter (size: 10 - 30 pixels; sigma: 10 - 20 pixels; pixel size: 6.5 mm). We then extracted from the smoothed time-lapse images the temporally varying brightness for each pixel. Within the imaging session of each moving direction, the data were smoothed temporally with a Gaussian filter (size: 5 frames; sigma: 3 frames; frame rate: 3 frames/s). With the moving direction of the dark bar being $\theta$ and the moving speed $v$, if at time $t$ the interrogated pixel reached its maximum calcium response $\Delta F/F$, the center of the receptive field $(x, y)$ for this pixel satisfied the equation:

$$x \cos\theta + y \sin\theta = v(t - 0.5T) \qquad (1)$$

where the origin of the coordinates was at the center of the bright background and $T$ was the time it took for the black bar moving from one end to the other. Although two orthogonal moving directions were sufficient to solve for $(x, y)$, we used eight directions ($\theta_1 = 0°, \theta_2 = 45°, \cdots, \theta_8 = 315°$) to increase robustness. Thus, for each

pixel we had the center of its corresponding receptive field:

$$A\begin{pmatrix} x \\ y \end{pmatrix} = B, \text{ where } A = \begin{pmatrix} \cos\theta_1 & \sin\theta_1 \\ \cos\theta_2 & \sin\theta_2 \\ \vdots & \vdots \\ \cos\theta_8 & \sin\theta_8 \end{pmatrix} \text{ and } B = \begin{pmatrix} t_1 - 0.5T \\ t_2 - 0.5T \\ \vdots \\ t_8 - 0.5T \end{pmatrix} \qquad (2)$$

We used the backslash operator for the least-squares solution of this overdetermined system:

$$\begin{pmatrix} x \\ y \end{pmatrix} = A \backslash B \qquad (3)$$

To exclude pixels whose receptive fields were not covered by the moving black bar, only pixels with $\Delta F/F > 10\%$ were used. The retinotopic map was rounded to a $3 \times 3$ color grid for visualization in Fig. 1c.

## Pixel-based OS and maximal response maps

To enhance the signal-to-noise ratio, we applied Gaussian smoothing (size = $3 \times 3$ pixels, sigma = 1 pixel) to raw images. We then extracted the fluorescence signal trace F for each pixel. $F_0$ was the averaged F values during the presentation of gray screen and used as the baseline fluorescence signal. The calcium transient trace $\Delta F/F$ was calculated as $(F-F_0)/F_0$. We obtained from $\Delta F/F$ the response $R$ of each pixel to a visual stimulus by averaging $\Delta F/F$ across the duration of drifting grating presentations. We then ran a one-way ANOVA test to find pixels whose responses across the 12 drifting directions were significantly different ($p < 0.05$), which we defined as the orientation-selective (OS) pixels. For the ANOVA tests here and below, the distribution of $R$ was assumed to be normal and variances were assumed to be equal across grating angle $\theta$ but this was not formally tested.

We then calculated $R(\theta)$, the trial-averaged response to drifting grating at $\theta$. The preferred grating drifting angles $\theta_{pref}$ for OS pixels were obtained by fitting the normalized $R(\theta)$ values with a bimodal Gaussian function:

$$R(\theta) = R_{offset} + R_{pref} e^{-\frac{\text{ang}(\theta - \theta_{pref})^2}{2\sigma^2}} + R_{oppo} e^{-\frac{\text{ang}(\theta - \theta_{pref} + 180)^2}{2\sigma^2}} \qquad (4)$$

where $R_{offset}$ is a constant offset, and $R_{pref}$ and $R_{oppo}$ are the responses at $\theta_{pref}$ and $\theta_{pref}$-180°, respectively. The function ang($x$) = min(|$x$|, |$x$−360|, |$x$+360|) wraps angular values onto the interval 0° to 180°. The OS pixels were assigned colors based on their preferred orientation (range: 0° to 180°, acquired by subtracting 180° from $\theta_{pref}$ if $\theta_{pref}$ was bigger than 180°) using a modified HSV colormap.

The maximal response of a pixel was defined as the maximum of $R(\theta)$ (i.e., the maximum of the trial-averaged responses towards the 12 drifting gratings) and used to generate the maximal response map.

## ROI extraction and orientation selectivity test

Elliptical regions of interest (ROIs) were drawn manually on the averaged images of calcium time series in ImageJ, which were then imported to MATLAB for extraction of fluorescence signal. Signal values for all pixels within a ROI were averaged for each frame to obtain the raw signal trace of the ROI, $F_{raw}$. For ROIs of neurons expressing cytosolic GCaMP6s, averaged fluorescent signal from neuropil area (from 5 to 10 pixels off the ROI border, 1.5 μm/pixel) was calculated as $F_{neuropil}$, the neuropil trace. $\Delta F_{neuropil}$ was calculated by subtracting $F_{0, neuropil}$ from $F_{neuropil}$, where $F_{0, neuropil}$ was the mean of the lowest 10% values in $F_{neuropil}$. Then $\Delta F_{neuropil}$ was multiplied by 0.7[86] and subtracted from $F_{raw}$, to remove neuropil contamination and generate

$F_{\text{true}}$. Calcium transient amplitude $\Delta F/F$ was calculated as $(F_{\text{true}} - F_{0,\text{true}})/F_{0,\text{true}}$. $F_{0,\text{true}}$ was the averaged $F_{\text{true}}$ value during the presentation of gray screen. For neurons expressing nucleus-targeted GCaMP6s, the neuropil subtraction step was skipped. The $\Delta F/F$ response to each visual stimulus was calculated as the average of all trials. A ROI was considered active if its maximal $\Delta F/F$ value was above 10%[28]. A ROI was considered to possess visually evoked activity if its activity during at least one visual stimulus was significantly higher than its activity during the inter-stimulus period (gray screen presentation) by paired $t$ test with $p < 0.01$. In full screen visual stimulation (no-edge cohort in Supplementary Fig. 6), there were 389 active (92.0%) out of total 423 cells, with 380 cells being both active and exhibiting visually evoked activity (98% of active cells).

To determine the orientation selectivity of individual neurons, we carried out the following tests. As in the pixel-based analysis, the response $R$ of each ROI to a visual stimulus was defined as the average $\Delta F/F$ across the period of drifting gratings. A ROI was defined as orientation selective if its responses across the various drifting directions were significantly different by one-way ANOVA test ($p < 0.05$) and if its tuning curve (normalized $R(\theta)$) is well fit by the bimodal Gaussian function described above. We evaluated the goodness of the fit with the fitting error $E$ and the coefficient of determination $\mathcal{R}^2$, calculated by the following equations:

$$E = \sum \left( R(\theta) - R_{\text{fitted}}(\theta) \right)^2 \tag{5}$$

$$\mathcal{R}^2 = 1 - \frac{\sum \left( R(\theta) - R_{\text{fitted}}(\theta) \right)^2}{\sum \left( R(\theta) - \bar{R} \right)^2} \tag{6}$$

Here $R(\theta)$ and $R_{\text{fitted}}(\theta)$ are the measured and fitted response at $\theta$, respectively. $\bar{R}$ is the mean of $R(\theta)$. Only ROIs with $E < 0.4$ and $\mathcal{R}^2 > 0.6$ were defined as orientation-selective (OS).

### ROI tuning parameter calculation

The orientation selectivity index (OSI), directional selectivity index (DSI), global OSI (gOSI), global DSI (gDSI), and tuning width (FWHM) was calculated according to our previous publication[28,87] following established definitions[18,20]. OSI was computed as $(R_{\text{pref}} - R_{\text{ortho}})/(R_{\text{pref}} + R_{\text{ortho}})$, with $R_{\text{pref}}$ and $R_{\text{ortho}}$ being the fitted responses at the preferred and orthogonal orientations, respectively. DSI was defined as $(R_{\text{pref}} - R_{\text{oppo}})/(R_{\text{pref}} + R_{\text{oppo}})$, with $R_{\text{pref}}$ and $R_{\text{oppo}}$ being the fitted response at the preferred and opposite directions, respectively. gOSI or gDSI was calculated as $\text{gOSI} = \frac{|\sum R(\theta)e^{i2\theta}|}{\sum R(\theta)}$ and $\text{gDSI} = \frac{|\sum R(\theta)e^{i\theta}|}{\sum R(\theta)}$. The tuning width for the preferred orientation is calculated as the full width at half maximum (FWHM) of the bimodal Gaussian function $2\sqrt{2\ln 2}\sigma$.

To determine whether neurons closer to one another had more similar orientation/direction preference within a FOV, we compared the distributions of the orientation/direction angle difference of neuron pairs that were nearest neighbors with those of randomly selected neuron pairs (Fig. 3) following a previous study. The angle difference between the preferred orientations of nearest neuron pairs was plotted as a cumulative distribution curve (red curves, right panels, Fig. 3c, i). To generate the cumulative distribution curves for random neuron pairs, for each neuron, we randomly select another neuron within the same FOV to calculate their preferred orientation difference, from which a cumulative distribution curve was generated (gray curves, right panels, Fig. 3c, i). This process was repeated 1000 times. Averaging the 1000 cumulative distribution curves gave rise to the combined randomized curve (blue curves, right panels, Fig. 3c, i). We then compared the nearest

neighbor distribution and the combined randomized distribution using two-sample Kolmogorov–Smirnov test.

### Statistics

Standard functions and custom-made scripts in MATLAB were used to perform all analysis. The data were tested for normal distribution using Kolmogorov–Smirnov test. Parametric tests were used for normally distributed data and non-parametric tests were applied to all other data. Kruskal–Wallis test is used for multigroup comparison in Fig. 2e–i. One-way ANOVA test with post-hoc Tukey's multiple comparison was used for the normally distributed data in Fig. 5c, e. Wilcoxon signed rank test was used to examine the paired data in Supplementary Fig. 8. In Fig. 6e, boxplots represented median and 25th–75th percentiles and their whiskers shown in Tukey style (plus or minus 1.5 times IQR). The statistical significance was defined as $^*p < 0.05$, $^{**}p < 0.01$, $^{***}p < 0.001$. Sample sizes were not predetermined by statistical methods but were based on those commonly used in the field.

### Reporting summary

Further information on research design is available in the Nature Portfolio Reporting Summary linked to this article.

## Data availability

Source data are provided with this paper. All other data are available from the lead contact, Na Ji (jina@berkeley.edu) upon request. Source data are provided with this paper.

## Code availability

Custom written MATLAB codes are available at https://github.com/JiLabUCBerkeley/SCimaging.

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

## Acknowledgements

We thank Li Zhaoping for helpful discussions, Wenzhi Sun for initializing SC surgical and imaging preparations, Marla Feller for critical comments on the manuscript. This work was supported by the Howard Hughes Medical Institute (N.J.), startup fund from the University of Maryland School of Medicine (Y.L.), the Weill Neurohub (N.J.), and National Institutes of Health (1U01NS120820 and U01NS118300) (N.J.).

## Author contributions

N.J. supervised the project; Y.L., R.L., K.B., and N.J. designed the experiments; Y.L. and K.B. carried out mouse surgeries and collected all data; R.L. developed visual stimulation and data analysis programs; Y.L., R.L., and K.B. analyzed data; Y.L., R.L., K.B., and N.J. prepared figures and wrote the manuscript.

## Competing interests

The authors declare no competing interests.
