## [Peer Review File · Nature Communications]

Stimulus edges induce orientation tuning in superior colliculusREVIEWER COMMENTS

Reviewer #1 (Remarks to the Author):

In this paper, the authors performed calcium imaging in the mouse SC and found that the edges of the stimulus could induce orientation-specific responses in SC neurons – stronger responses to the orientations perpendicular to the edge. The experiments were done with high quality and the findings interesting. All my comments below are minor and meant to improve the presentation.

1. The main finding could resolve some of the controversies regarding whether there are columns in the mouse SC – previous reports of orientation/direction columns are likely just artifacts of an edge effect. I suggest that the authors highlight this point by adding plots to figure 3 and/or 4, illustrating the lack of columns when analyzing neurons not on the edges.
2. Some statistical tests are needed for Fig 3. The clustering of OS preference seems obvious, especially for deeper cells. But the same is not obvious for DS. Statements like “DS neurons within the imaging FOVs exhibited clustering” should be supported by statistics.
3. The observation that DS cells were unlikely to switch is interesting (Fig. 4H) and deserves more analyses. I suggest, in addition to the population plots already in shown, some scatter plots of individual neurons. Maybe a plot of the change in direction preference vs. the neuron’s gDSI. Probably similar plots for OS as well, to illustrate the difference between the two groups of cells. Such plots and maybe others could be helpful to illustrate what predicts the switch (depending on their selectivity, preference, etc), thereby suggesting underlying mechanisms.
4. Is the edge effect specific to the SC? I am NOT asking the authors to perform new experiments on this, but it would be ideal if they have some data from V1 already (or retina), or can at least speculate on this.
5. The findings in Fig. 5 are reminiscent of and consistent with Barchini et al, 2018, eLife. That paper should be cited.

Reviewer #2 (Remarks to the Author):

The authors of this study explore the functional organization of the visually responsive layers of the mouse superior colliculus (SC) with a stimulus consisting of moving oriented gratings that are retinotopically limited to a window of a specific shape, most often a circle, or square. They find that the retinotopic region of the superior colliculus that represents the boundary of the grating stimulus window exhibits the strongest responses and these responses exhibit a preference for grating orientation (and in some cases direction) that varies as a function of retinotopic position along the

boundary. They also demonstrate that the orientation preference of the SC neurons varies as a function of the grating size and shape within the window and they interpret this as both different from the organization for orientation selectivity found in visual cortex of primate and cat, and as a reflection of a sensitivity of SC neurons to stimulus saliency.

Major Concern:

While there is no doubt that the authors have provided a detailed account of a phenomenon they have encountered in the mouse SC, they do not appear to be aware of a sizeable literature on the response properties of orientation selective neurons in the visual cortex that would be entirely consistent with these observations: the representation of stimulus spatiotemporal energy.

Adelson and Bergen J. Opt. Soc. Am. 1985. A 2, 284–299

Carandini et al., 1999. Cerebral Cortex 13:401-443

Basole et al., 2003. Nature 423: 986-990

Basole et al., 2006. Progress in Brain Research 154: 121-134

Mante, V. and Carandini, M., 2003. Current Biology, 13(23), pp. 906-908.

Mante, V. and Carandini, M., 2005. Journal of Neurophysiology, 94(1), pp.788-798

Mante, V. and Carandini, M., 2010. J. Neurosci, 30(6), pp.1985-1993

The authors appear not to appreciate the complex nature of the stimulus that has been used in these experiments. Its not just a uniform grating stimulus, but line segments with end points that possess energy at many orientations in the Fourier domain that then interact with the movement of the stimulus. Serious attention needs to be given to the spatiotemporal energy framework for thinking about these results, and what this means about the overall significance of the findings. As it stands, the claim that this is somehow different than what has been described in visual cortex is not accurate.

We thank both reviewers for their helpful comments and suggestions, which have helped us improve the manuscript. We have included new data and analysis and made point-by-point responses below and have indicated where edits were made in the revised manuscript (red fonts below; line numbers refer to those in the revised manuscript with all edits accepted – a change-tracked file “Manuscript_tracked_change.pdf” is also provided).

Reviewer #1:

In this paper, the authors performed calcium imaging in the mouse SC and found that the edges of the stimulus could induce orientation-specific responses in SC neurons – stronger responses to the orientations perpendicular to the edge. The experiments were done with high quality and the findings interesting. All my comments below are minor and meant to improve the presentation.

1. The main finding could resolve some of the controversies regarding whether there are columns in the mouse SC – previous reports of orientation/direction columns are likely just artifacts of an edge effect. I suggest that the authors highlight this point by adding plots to figure 3 and/or 4, illustrating the lack of columns when analyzing neurons not on the edges.

We carried out the analysis as recommended by the reviewer. In Figure S4 (see below), we contrasted the spatial organizations of OS neurons located away from the stimulus edge (“Off-edge”) and on the stimulus edge (“Edge”).

Figure S4. Lack of orientation clustering away from edge.

(A,D) sSC neurons expressing nuclear-targeted GCaMP6s at different depths in a cortex-removed mouse. (B,E) OS neurons color-coded by their preferred orientation. Dashed line: approximate location of grating edge. (C,C',F,F') Left: Cumulative distributions of preferred orientation difference between nearest neighbors (red curve), randomly selected neuron pairs (1,000 repeats, gray curves), and the average of random distributions (blue curve); Right: Difference in preferred orientations for pairs of OS neurons versus their horizontal separation (red curve: averages of 50- μ m bins), for neurons (C,F) away from or (C',F') near stimulus edge. p value: two-sample Kolmogorov-Smirnov test between red and blue curves. Representative result from N = 3 mice. (G,H,I) Similar analysis to (A,B,C) for neurons expressing GCaMP6s cytosolically. Representative result from N = 4 mice.

We added to the manuscript (Lines 132-133): “Furthermore, analysis on FOVs retinotopically mapped away from the stimulus edges indicated a lack of orientation clustering (Figure S4).”

2. Some statistical tests are needed for Fig 3. The clustering of OS preference seems obvious, especially for deeper cells. But the same is not obvious for DS. Statements like “DS neurons within the imaging FOVs exhibited clustering” should be supported by statistics.

Thank you for the suggestion. Following Chen et al., J Neuroscience (2021) 41, 461, we carried out statistical analysis of the clustering for both OS and DS. Consistent with our previous conclusions, we observed statistically significant clustering in directional preferences. We have updated Fig. 3:

Figure 3. Stimulus edge leads to orientation and direction columns in both cortex-intact and cortex-removed mice. (A) sSC neurons expressing nuclear-targeted GCaMP6s at different depths in a cortex-intact mouse. (B) OS neurons mapped to the edge of 80° circular drifting gratings, color-coded by their preferred orientation. (C) Left: Difference in preferred orientations for pairs of OS neurons versus their horizontal separation (red curve: averages of 25-μm bins). Right: Cumulative distributions of preferred orientation difference between nearest neighbors (red curve), randomly selected neuron pairs (1,000 repeats, gray curves), and the average of random distributions (blue curve). P value: two-sample Kolmogorov-Smirnov test between red and blue curves. (D) DS neurons with their preferred directions of motion indicated by arrows. (E) Difference in preferred directions for pairs of DS neurons versus their horizontal separation. (F) Left: Difference in preferred directions wrapped to (0, 90°) for pairs of DS neurons versus their horizontal separation; Right: same analysis as in (C). Representative result from 5 mice. (G, H, I, J, K, L) Same as (A, B, C, D, E, F), respectively, but acquired from a mouse with overlaying cortex removed. Representative result from 3 mice.

We added to the manuscript (Lines 119 – 121): “Comparing the preferred orientation angle difference of neuron pairs that were nearest neighbors with those of randomly selected neuron pairs, we found that the nearest neighbors had significantly more similar orientations (two-sample Kolmogorov-Smirnov test).”

Lines (126-129): “Wrapping the preferred direction difference to $(0, 90^\circ)$ (i.e., if direction difference is larger than 90° , use its supplementary angle), we found this clustering of similar/opposite direction preference of nearby neurons to be statistically significant (Figure 3F; two-sample Kolmogorov-Smirnov test).”

3. The observation that DS cells were unlikely to switch is interesting (Fig. 4H) and deserves more analyses. I suggest, in addition to the population plots already in shown, some scatter plots of individual neurons. Maybe a plot of the change in direction preference vs. the neuron’s gDSI. Probably similar plots for OS as well, to illustrate the difference between the two groups of cells. Such plots and maybe others could be helpful to illustrate what predicts the switch (depending on their selectivity, preference, etc), thereby suggesting underlying mechanisms.

We thank the reviewer for this suggestion. We have generated and included 4 scatter plots in the updated Fig. 4:

For each AS neuron, we plotted its two gOSI values vs. its preferred orientations for gratings with horizontal or vertical edge:

For each AS neuron, we plotted its two gOSI values for gratings with horizontal or vertical edge vs. the difference in its preferred orientations for gratings with horizontal or vertical edge:

For each DS neuron, we plotted its two gDSI values vs. its preferred orientations for gratings with horizontal or vertical edge:

For each DS neuron, we plotted its two gDSI values for gratings with horizontal or vertical edge vs. the difference in its preferred directions for gratings with horizontal or vertical edge:

These plots indicate that:

1. The selectivity of the AS and DS neurons (i.e., gOSI and gDSI distributions) does not vary with the edge orientations – this is consistent with the histograms for gOSI, gDSI, and FWHM of the tuning curves.
2. For AS/DS neurons, there is no obvious correlations between the degree of preferred orientation/direction difference and its gOSI/gDSI.

We have incorporated these panels to the revised Fig. 4G-J, and added the following sentences (Lines 175 – 177):

“These properties appeared to be independent of how orientation-selective (for AS neurons) or direction-selective (for DS neurons) these neurons were, as neurons with large or small changes in their preferred orientation/direction had their gOSI/gDSI values within similar ranges (Fig. 4H,J).”

4. Is the edge effect specific to the SC? I am NOT asking the authors to perform new experiments on this, but it would be ideal if they have some data from V1 already (or retina), or can at least speculate on this.

We indeed have data from V1. As shown below for small versus large circular grating, we do not see similar edge effect in L2/3 neurons of V1.

Mouse primary visual cortex lacks similar edge-induced effect. L2/3 neurons labeled with AAV2/1.syn.GCaMP6s in left V1 responding to circular drifting gratings of (A-C) 25° or (D-F) 50° view angles. (A,D) Maximal calcium responses across all orientations. (B,E) Maps of (left) OS pixels and (right) OS neurons, color-coded by their preferred orientation. (C,F) Left: Difference in preferred orientations for pairs of OS neurons versus their horizontal separation (red curve: averages of 50- μ m bins). Right: Cumulative distributions of preferred orientation difference between nearest neighbors (red curve), randomly selected neuron pairs (1,000 repeats, gray curves), and the average of random distributions (blue curve). p value: two-sample Kolmogorov-Smirnov test between red and blue curves.

We added to Discussion (Lines 283-284): “In *in vivo* calcium imaging experiments in L2/3 neurons of the mouse V1, we did not observe similar edge-induced effect (data not shown).”

Regarding RGCs, we already have the following in Discussion: “the possibility that saliency computation starts in retina remains. Given the existence of similar center-surround interactions and suppression by stimuli in the extra-classical surround in the retina, saliency mapping may already exist at the retinal output, which can be investigated by imaging the axonal inputs from the retina in sSC.”

We are currently performing similar experiments in retina and will report the results in a separate manuscript. Briefly, we do see similar effects in RGCs.

5. The findings in Fig. 5 are reminiscent of and consistent with Barchini et al, 2018, eLife. That paper should be cited.

We thank the reviewer for pointing out this reference. We have now cited as Ref. 61.

Reviewer #2 (Remarks to the Author):

The authors of this study explore the functional organization of the visually responsive layers of the mouse superior colliculus (SC) with a stimulus consisting of moving oriented gratings that are retinotopically limited to a window of a specific shape, most often a circle, or square. They find that the retinotopic region of the superior colliculus that represents the boundary of the grating stimulus window exhibits the strongest responses and these responses exhibit a preference for grating orientation (and in some cases direction) that varies as a function of retinotopic position along the boundary. They also demonstrate that the orientation preference of the SC neurons varies as a function of the grating size and shape within the window and they interpret this as both different from the organization for orientation selectivity found in visual cortex of primate and cat, and as a reflection of a

sensitivity of SC neurons to stimulus saliency.

Major Concern:

While there is no doubt that the authors have provided a detailed account of a phenomenon they have encountered in the mouse SC, they do not appear to be aware of a sizeable literature on the response properties of orientation selective neurons in the visual cortex that would be entirely consistent with these observations: the representation of stimulus spatiotemporal energy.

Adelson and Bergen J. Opt. Soc. Am. 1985. A 2, 284–299

Carandini et al., 1999. Cerebral Cortex 13:401-443

Basole et al., 2003. Nature 423: 986-990

Basole et al., 2006. Progress in Brain Research 154: 121-134

Mante, V. and Carandini, M., 2003. Current Biology, 13(23), pp. 906-908.

Mante, V. and Carandini, M., 2005. Journal of Neurophysiology, 94(1), pp.788-798

Mante, V. and Carandini, M., 2010. J. Neurosci, 30(6), pp.1985-1993

The authors appear not to appreciate the complex nature of the stimulus that has been used in these experiments. Its not just a uniform grating stimulus, but line segments with end points that possess energy at many orientations in the Fourier domain that then interact with the movement of the stimulus. Serious attention needs to be given to the spatiotemporal energy framework for thinking about these results, and what this means about the overall significance of the findings. As it stands, the claim that this is somehow different than what has been described in visual cortex is not accurate.

We thank the reviewer for referring us to the papers on the stimulus Spatiotemporal Energy Model and the experiment carried out in the Ferret V1 by Basole 2003. We have carried out additional experiments using stimuli similar to those in Basole 2003. We have also thought through how this model may explain our experimental results.

We have included a new figure and discussion in the revised manuscript, as detailed below. With the additional experiments, we believe that the main thesis of our manuscript, i.e., our discovery of “a columnar organization of stimulus-dependent orientation selectivity in the early processing of visual information” in the mouse superior colliculus, remains valid.

New discussion:

The stimulus spatiotemporal energy model can explain the population tuning properties that we observed in SC. We have added references to this model and a new paragraph to our Discussion (Lines 264 – 278):

“Our data on the population activity of the sSC neurons can be understood within the framework^{56, 57, 58, 66} of the spatiotemporal energy model (SEM). In SEM, the spatiotemporal receptive field of a neuron in space-time (x,y,t) can also be described in the spatial and temporal frequency domain (w_x, w_y, w_t) . To determine how neurons respond to a specific visual stimulus, one looks at the spatiotemporal frequency domain representation of the stimulus. Neurons with receptive fields inside the volume occupied by the visual stimulus in the spatiotemporal frequency domain are activated by the stimulus. Therefore, SEM predicts that a stimulus with a larger volume in the spatiotemporal frequency space elicits responses from a larger population of neurons. This is consistent with our maximal response map analysis (Figs. 1D, S1A, 2B, 6B), where the strongest population responses were located at the edge of our stimuli. Discontinuities in (x,y) at the grating edge lead to a larger volume in the frequency domain and thus evoke responses from more neurons. SEM can also explain our observation of heightened activity at the salient boundaries between distinct visual stimuli: The salient boundaries are

discontinuous by definition, and thus activate more neurons. SEM is also successful in explaining the increase of response magnitude with the increase in the spatial frequency of grating stimuli (Fig. S3B): gratings with higher spatial frequency are composed of narrower line segments, the ends of which occupy a larger volume in (wx,wy,wt). These gratings thus elicit responses from more neurons and lead to a larger maximal response, as we observed experimentally.”

New data section (Line 180-211):

“Edge-induced changes in orientation selectivity of sSC neurons are distinct from those induced by moving texture stimuli

Figure 5. Orientation tuning of sSC neurons in response to moving texture stimuli. (A) A representative FOV of sSC neurons with the OS neurons color-coded by their preferred grating/bar orientation for grating/moving bar stimulus sets. (B-E) Histogram distributions of preferred orientation angles for neurons that are orientation selective towards (B) grating, (C) ortho-bars, (D) clockwise-bars, and (E) counterclockwise-bars stimulus sets, respectively. (F) Histogram distribution of difference in preferred orientation angles for neurons that were OS for both grating and ortho-bars stimulus sets. (G) Histogram distribution of difference in preferred orientation angles for neurons that were OS for ortho-bars & clockwise-bars, ortho-bars & counterclockwise, or clockwise-bars & counterclockwise stimulus sets.

In addition to drifting gratings moving in directions orthogonal to grating orientation, we also measured the responses of sSC neurons towards three sets of texture stimuli composed of oriented moving bars. These experiments were inspired by previous work in the ferret V1, where changing the axis of motion of such stimuli resulted in striking shifts in the population orientation-tuning of V1 neurons⁵⁵. We asked whether the same population tuning shifts happen in sSC, and if they do, how the population tuning shifts manifest themselves in tuning of individual sSC neurons.

We imaged at single-cell resolution the responses of neurons expressing cytosolic GCaMP6s in sSC exposed by cortical removal towards four sets of stimuli extending over 30° visual field (10× 0.6-NA or 16× 0.8-NA

objective, 7 FOVs in 4 mice, N=2,204 visually responsive neurons; Fig. 5). They included gratings moving along a direction orthogonal to the grating orientation (“grating”; 0.04 cycles/degree, 1.5 Hz), bars moving in a direction orthogonal to the orientation of the bars (“ortho-bars”; 10° in length, same speed as grating), and bars moving obliquely with the bar orientation being 45° clockwise (“clockwise-bars”) or 45° counter-clockwise (“counterclockwise-bars”) from its motion direction. From the calcium responses of individual neurons, we determined the preferred orientations of individual sSC neurons (Fig. 5A).

We first investigated the population orientation tuning elicited by these stimulus sets. We found that grating and ortho-bars stimuli led to similar population tuning as indicated by the histogram distributions of preferred orientations (Fig. 5B,C). But when we changed the axis of motion of the bar stimuli from orthogonal to oblique relative to the bar orientation, we observed shifts in population tuning (Fig. 5D,E). These observations are consistent with previous work⁵⁵ in V1 and may be similarly explained by the stimulus spatiotemporal energy model^{56, 57, 58}.

We then evaluated how tuning changes for individual neurons, by identifying neurons that were OS in pairs of stimulus sets and investigating how they changed their preferred orientation angles. For individual neurons that were OS for both grating and ortho-bars, most did not change their preferred orientation (Fig. 5F). For neurons that were OS for two bar stimulus sets, a higher fraction of neurons exhibited a substantial change in their preferred orientation angle (Fig. 5G). However, because only a minority of OS neurons tuned in one bar stimulus set were also OS in another bar stimulus set (0.33 ± 0.09 , mean \pm S.D.), the fraction of OS neurons for a specific stimulus set that changed their preferred orientation in response to changes in motion of these texture stimuli was low. In other words, the observed population tuning changes were mostly generated by different populations of neurons that became tuned for different stimulus sets. This is distinctively different from the orientation preference changes induced by grating edges described in the previous section, where a majority of OS neurons changed their orientation preference in response to changes in edge orientation (Fig. 4H).”

In summary:

For moving texture stimuli, ~33% of neurons were OS for a pair of stimulus sets (e.g., Ortho-bars versus Clockwise-bars). For these OS neurons, the change of their preferred orientations has the following distribution:

For edged-grating stimuli, >90% of neurons were OS for the pair of stimulus sets (i.e., gratings with a vertical edge versus gratings with a horizontal edge, Lines 161-162). For these OS neurons, the change of their preferred orientations has the following distribution:

In other words, our data indicate that although the stimulus spatiotemporal energy model explains the shifts in population activity, using moving texture stimuli does not lead to the orientation tuning changes for individual sSC neurons that we observed for grating edges. Hence our results for individual neurons suggest that additional mechanisms underlie the tuning changes at grating edges.

What is the mechanism that explains this edge-induced tuning change in sSC? Since our manuscript is focused on reporting a novel experimental observation, we did not include a detailed theoretical model here. However, we are developing a surround-suppression based (“difference of Gaussian”) model that can explain the edge-induced tuning change. Since the submission of our original manuscript, we have extended our investigation of the same effects in the retinal ganglion cells. We are working on a manuscript that will detail our theoretical model and how it applies to both sSC data and the new data from RGCs.

REVIEWERS' COMMENTS

Reviewer #1 (Remarks to the Author):

The authors should be commended for the extensive revisions to address previous reviews. The newly added data and analysis have addressed all my concerns and significantly improved the manuscript. I have only one minor point for the authors to consider -

It's about the new Fig. 3E,F, K and L, and associated text in lines 126-129: "Wrapping the preferred direction difference to $(0, 90^\circ)$ (i.e., if direction difference is larger than 90° , use its supplementary angle), we found this clustering of similar/opposite direction preference of nearby neurons to be statistically significant (Figure 3F; two-sample Kolmogorov-Smirnov test)."

These statements are not wrong, but the message is confusing and potentially misleading, due to the "wrapping" of direction difference. By doing that, you really are looking at motion axis, instead of directions. I suggest some text change in lines 124-126, perhaps like the following, "In contrast to the OS neurons, DS neurons within the FOV do not exhibit simple/straightforward clustering of their preferred directions (Fig. 3E). Instead, nearby DS neurons tend to prefer similar or opposite directions of the same motion axis (Figure 3E). Wrapping the preferred ..." .

Reviewer #2 (Remarks to the Author):

The authors have now addressed the concerns about the complexity of the stimulus configuration and placed their observations in the context of previous studies in visual cortex. This significantly improves the interpretation of the result, and the authors have identified additional features of the SC response properties that will be the subject of future studies.

REVIEWERS' COMMENTS

Reviewer #1 (Remarks to the Author):

The authors should be commended for the extensive revisions to address previous reviews. The newly added data and analysis have addressed all my concerns and significantly improved the manuscript. I have only one minor point for the authors to consider -

It's about the new Fig. 3E,F, K and L, and associated text in lines 126-129: "Wrapping the preferred direction difference to $(0, 90^\circ)$ (i.e., if direction difference is larger than 90° , use its supplementary angle), we found this clustering of similar/opposite direction preference of nearby neurons to be statistically significant (Figure 3F; two-sample Kolmogorov-Smirnov test)."

These statements are not wrong, but the message is confusing and potentially misleading, due to the "wrapping" of direction difference. By doing that, you really are looking at motion axis, instead of directions. I suggest some text change in lines 124-126, perhaps like the following, "In contrast to the OS neurons, DS neurons within the FOV do not exhibit simple/straightforward clustering of their preferred directions (Fig. 3E). Instead, nearby DS neurons tend to prefer similar or opposite directions of the same motion axis (Figure 3E). Wrapping the preferred ..." .

Response: We thank the reviewer for the compliment and agree with the recommended changes. The manuscript has been revised using the edit suggestion by the reviewer.

Reviewer #2 (Remarks to the Author):

The authors have now addressed the concerns about the complexity of the stimulus configuration and placed their observations in the context of previous studies in visual cortex. This significantly improves the interpretation of the result, and the authors have identified additional features of the SC response properties that will be the subject of future studies.

Response: We sincerely appreciate the thoughtful and detailed feedback from the reviewer, which has significantly contributed to the improvement of our manuscript.